# Development and Evaluation of Correction Models for a Low-Cost Fine Particulate Matter Monitor

Brayden Nilson[12], Peter L. Jackson[1], Corinne L. Schiller[12], Matthew T. Parsons[2]

[1]Department of Geography, Earth and Environmental Sciences, University of Northern British Columbia, Prince George, V2N 4Z9, Canada

[2]Air Quality Science – West, Meteorological Service of Canada, Environment and Climate Change Canada, Vancouver, V6C 3S5, Canada

*Correspondence to*: Brayden Nilson (brayden.nilson@ec.gc.ca)

**Abstract.** Four correction models with differing forms were developed on a training data set of 32 PurpleAir-Federal Equivalent Method (FEM) hourly fine particulate matter (PM$_{2.5}$) observation colocation sites across North America (NA). These were evaluated in comparison with four existing models from external sources using the data from 15 additional NA colocation sites. Colocation sites were determined automatically based on proximity and a novel quality control process. The Canadian AQHI+ system was used to make comparisons across the range of concentrations common to NA, as well as to provide operational and health-related context to the evaluations. The model found to perform the best was our Model 2, PM$_{2.5\text{-corrected}}$ = PM$_{2.5\text{-cf-1}}$ / (1 + 0.24 / (100 / RH$_\%$ - 1)) – RH is limited to the range [30 %, 70 %], which is based on the RH growth model developed by Crilley et al. (2018). Corrected concentrations from this model in the moderate to high range, the range most impactful to human health, outperformed all other models in most comparisons. Model 7 (Barkjohn et al., 2021) was a close runner up and excelled in the low concentration range (most common to NA). The correction models do not perform the same at different locations, so we recommend testing several models at nearby colocation sites and utilizing that which performs best if possible. If no nearby colocation site is available, then we recommend using our Model 2. This study provides a robust framework for the evaluation of low-cost PM2.5 sensor correction models and presents an optimized correction model for North American PA sensors.

**Keywords:** North America, air quality, PM$_{2.5}$, low-cost sensors, colocation, corrections

## 1 Introduction

Fine particulate matter (PM$_{2.5}$) is a primary air pollutant of concern for human health in Canada and the USA (EPA, 2021; Government of Canada, 2021) as well as world-wide. PM$_{2.5}$ consists of suspended aerosols with diameters ≤ 2.5 μm which when inhaled can enter the lower respiratory tract and potentially pass through into the blood (Feng et al., 2016). PM$_{2.5}$ can have high spatial and temporal variability and impact highly populated areas (Li et al., 2019). Globally, concentrations of PM$_{2.5}$ are highest in South Asia, Africa, and the Middle East, while North America has some of the lowest concentrations on average (Health Effects Institute, 2020). Smoke from wildfires is increasingly becoming an issue for PM$_{2.5}$ concentrations in North America and other parts of the world. Both chronic and acute exposure to PM$_{2.5}$ have been linked to a higher risk of mortality from cardiovascular and respiratory diseases (Lelieveld et al., 2015; Mcguinn et al., 2017; Bowe et al., 2019).

Near-real-time observations of PM$_{2.5}$ have historically been restricted by cost and access. As a result, many cities rely only on a single PM$_{2.5}$ monitor (or none at all) provided by governmental networks. These Federal Equivalent Method (FEM) monitors are expensive, require a large shelter, and require regular maintenance to provide accurate and reliable data. However, rigorous testing and standardization ensures FEM observations are equivalent to reference monitoring methods. Recently, a growing interest in low-cost/small sensor technology has allowed for a rapid expansion of observation networks. However, these low-cost

sensors are not as accurate as the FEM monitors, necessitating additional processing and testing before knowing how to interpret
their data and understanding how they may be used (Datta et. al, 2020; Zamora et. al 2019).

FEM monitors can be one of multiple potential sensor types; each of which measures $PM_{2.5}$ using different techniques (such as beta attenuation, gravimetric, and/or broadband spectroscopy). These sensors are rigorously tested and compared with 24-hour average reference measurements from Federal Reference Monitors (FRM) to ensure comparability. FEM monitors are in well-controlled shelters with incoming air being dried to minimise any humidity impacts.

The North American (NA) network of FEM monitors for $PM_{2.5}$ and other air pollutants contribute to the Canadian Air Quality Health Index (AQHI) system (for monitors in Canada) and the US Air Quality Index (AQI); both used to provide health advisories for periods of unhealthy levels of air pollution. The Canadian AQHI system is based on rolling three-hour averages of three pollutants: $PM_{2.5}$, $NO_2$ and $O_3$ (Stieb et al. 2008). The AQHI+ system is a simplification of AQHI which relies only on hourly-averaged $PM_{2.5}$ and shares the index value and associated health messaging (BC Lung Association, 2019). In British

Columbia, the AQHI+ overrides the regular AQHI formulation during events with high concentrations of $PM_{2.5}$, such as wildfire smoke conditions. The AQHI+ can consequently be used to facilitate direct health-related comparisons with low-cost sensors, which currently do not detect the other criteria air pollutants sufficiently (BC Lung Association, 2019). This system is based on the Canadian AQHI and was initially developed for British Columbia (but is now used mostly across Canada) to provide more accurate health messaging during wildfire smoke conditions (for which $PM_{2.5}$ is a main concern). AQHI+ ranges from 1 to 11,

with the last value represented as '10+' (Eq. 1).

$$AQHI+ = \ ceiling\left(\frac{PM_{2.5}}{10}\right) \ domain: [1, 11] \tag{1}$$

Low-cost sensor technology has been developed in recent years that can be used to provide increased spatial and temporal resolution of $PM_{2.5}$ measurements. In this study, low-cost sensors from PurpleAir (2021) which use two nephelometers to detect $PM_{2.5}$ were compared with colocated FEM monitors. A nephelometer estimates the particulate concentration by measuring light

from a source that is scattered by particles; concentrations are derived by correlating this scattering amplitude with a mass-based monitor (Hagan and Kroll, 2020). The relationship between scattering and particulate concentration is affected by particle refractive index, shape, density, and size distribution: nephelometers are calibrated for a known particulate. Error is introduced when the actual particle characteristics are different from those used for nephelometer calibration. $PM_{2.5}$ measured by these sensors are impacted by humidity, as liquid water can condense on the particles which may be interpreted as larger/more

particles by the sensor. FEM monitors handle this issue by using an internal humidity probe and either an inlet heater or dryer to control the relative humidity (British Columbia Ministry of Environment, 2020; Peters et. al 2008). These methods are not feasible for low-cost sensors, and therefore humidity effects as well as other biases that may exist need to be corrected. Multiple correction models are being produced by different groups to minimise these errors; many of which utilize RH in a correction formula (e.g., Ardon-Dryer et al. 2019, Barkjohn et al. 2021).

PurpleAir (PA) monitors are targeted to citizen scientists and air quality professionals alike as small, low-cost, easy-to-install devices and as such, have proliferated to form a global network of sensors with thousands of devices in NA alone. These monitors contain two Plantower PMS5003 nephelometers which each detect $PM_{2.5}$ (named "A" and "B"), as well as separate sensors for measuring RH and temperature. $PM_{2.5}$ concentration is reported by the sensors using two different proprietary correction factors ("$PM_{2.5}$ CF 1" and "$PM_{2.5}$ CF ATM") which convert the measured particle scattering amplitudes into the

reported concentrations. The "CF ATM" correction factor is derived from Beijing atmospheric conditions while "CF 1" was derived from a lab study using symmetrical particles of a known size and is recommended for use in industrial settings (Zhou, 2016; Yang, personal communication, 2019). The "CF 1" data were found to correlate better with FEM observations in our data

set. PM$_{2.5}$ concentrations from the PA monitors have shown promising results when a colocation study derived correction model is applied but tend to overestimate FEM readings otherwise (Kim et al., 2019; Malings et al., 2019; Li et al., 2020; Feenstra et al., 2020; Tryner et al., 2020).

Issues of comparability arise as these low-cost monitors typically rely on light scattering based detection alone. This method is inherently less reliable than most FEM monitoring methods (Mehadi et al., 2020). Humidity has a large impact on the accuracy of the PA observations since these low-cost sensors do not purposely dry the air (some lowering of RH likely occurs from the heat produced by the electronics). At RH levels around 50 % and higher, low-cost PM$_{2.5}$ monitors can start counting aerosol water as larger particles as water accumulates on the particles and impacts the accuracy of light scattering based methods (Jayaratne et al., 2018; Magi et al., 2019; Zamora et al., 2019). In addition, PA temperature observations tend to be biased high (and in turn RH biased low) because of internal heat produced by the electronics as well as incoming solar radiation (which has varying impacts depending on the physical location and placement of each monitor). However, the PM$_{2.5}$ observations still correlate well with gravimetric (Tryner et al., 2020), broadband spectroscopy (Kim et al., 2019; Li et al., 2020) and beta attenuation monitors (Zheng et al., 2018; Kim et al., 2019; Magi et al., 2019; Mehadi et al., 2020; Si et al., 2020), allowing for a correction model to be developed and applied.

Many correction models have already been developed for PA monitors, several of which have been selected here for comparison. Caution is advised when utilizing generalised models such as these, as they will not provide the same degree of improvement at all locations given differences in aerosol properties that a nephelometer cannot detect or differentiate. The US Environmental Protection Agency (EPA) recently released a multiple linear regression correction model which includes a RH term and is derived from 24-hour average data (Barkjohn et al., 2021). Several simple linear corrections are detailed on the PA mapping site (PurpleAir, 2021) including ones from LRAPA (2019) and Kelly et al. (2019). A recent study by Ardon Dryer et al. (2019) produced multiple corrections at different colocation sites in the US. We selected their multiple linear model (including both temperature and RH) for the average of sensors at their Salt Lake City site. We found this correction to perform best for our dataset in comparison with the others provided in the study.

In this study, a novel method for detecting and preparing PA-FEM monitor colocations sites across Canada and the United States is developed and implemented. Using the resulting dataset, multiple PA PM$_{2.5}$ correction models are developed and evaluated in comparison with those from outside parties. The correction which best improves the moderate to high concentration range (30 µg m$^{-3}$–100 µg m$^{-3}$) across multiple sites is selected as performing best in relation to health interpretations for general use purposes. It is our objective to create a correction model for general application across multiple sensors/locations, however, a more specialised correction is recommended where nearby colocation data are available.

## 2 Methods

The R statistical software (R Core Team, 2020) was used for data collection and analyses in this study. The R packages dplyr (Wickham et al. 2021), lubridate (Grolemund and Wickham, 2011), and stringr (Wickham, 2019) were used for data manipulation. Figures and maps were built using the ggplot2 (Wickham, 2016), ggmap (Kahle and Wickham, 2013), and leaflet (Cheng et al. 2021) packages.

### 2.1 Colocation Site Selection and Data Retrieval

An automated algorithm to detect potential PA/FEM monitor colocation sites and apply quality control (QC) methods was developed to identify the sites that were colocated, defined as one or more outdoor PA monitors being within 50 m of each other

(as of November 2020), and remove any periods of invalid data. Monitor coordinates were provided through the AirNow (for FEM monitors) and PA databases. Based on this, 86 sites were identified; however, further analysis was necessary to remove sites or periods of time where the PA monitors were likely not colocated or were located indoors. FEM monitor detector types were retrieved from the US AQS database (EPA, 2020) for the US stations; Canadian station information was collected through contact with the monitor operators.

FEM observation data were obtained from the AirNow database (AirNow, 2021) which provided hourly $PM_{2.5}$ concentration observations from sites across North America. PA data was retrieved from their ThingSpeak repository as hourly averages for comparison with the FEM monitors (PurpleAir 2021). Sensor A and sensor B "CF=1" data from each PA monitor were averaged together to produce a single record. Historically, the "CF=1" and "CF=ATM" were erroneously mislabelled in the PA data; this has since been resolved and it was ensured that the actual "CF=1" data were being used here.

## 2.2 Quality Control

Once the sites were chosen, the observation data were flagged as invalid using both automated tests and manual inspection. Emphasis was placed on using the data available from the PA monitor only as much as possible in order to ensure these QC methods would still be applicable at sites without a collocated FEM monitor. However, the final step of the QC process used the FEM data as well as those from PA to calculate the overall $PM_{2.5}$ correlation; this was to ensure the automated collocation site

detection did not inadvertently select a non-collocated monitor.

The first automated flagging did internal comparisons between the sensor A and sensor B in each PA monitor to identify failures from a particular sensor within the monitor (assuming both sensors have not failed in a similar way). Hours were flagged invalid where data from either A or B were missing, or where the absolute error between A and B for that hour was both greater than 5 $\mu g\ m^{-3}$ and greater than half of the mean of A and B (derived from similar methods released by Tryner et al., 2020 and Barkjohn

et al. 2021).

An automated flag was also developed to identify when a sensor was indoors rather than outside based on the temperature and RH daily profiles. PA monitors measure relative humidity (RH) and temperature, we used these observations to flag entire months as invalid based on the variability of those data within each month. Individual months were flagged as invalid where the standard deviation in both temperature and RH for that month were less than 3 units (i.e. 3 °C for temperature or 3% for RH) or

140 were identified as abnormally low using the Hampel Identifier with the standard cutoff of three median absolute deviations from the median, a commonly used outlier detection method for small sensors (Pearson 2002). Any month with less than 72 hours of data was also marked as invalid.

Manual inspections identified several individual hours for specific PA and FEM monitors that were marked invalid due to abnormally high concentrations reported. Data from AirNow have not been QC'ed by the responsible authorities, necessitating

this check. In addition to this, any PA monitors with less than 2 months of valid data were dropped from the analysis; most of these had sporadic or very poorly correlated observations.

Sites with multiple colocated PA monitors were averaged together to produce a single data record for each site after flagging and removing any invalid data. Colocation sites with less than half a year of valid colocation data were then removed to ensure representative data coverage from each site. We further removed several sites after viewing scatter plots of their valid PA and

150 FEM $PM_{2.5}$ observations and noticing a non-linear relationship quite different from other sites. The final set of colocation sites (47 in total) were then selected as those with at least half a year (4380 hours) of valid data from both PA and FEM and a minimum correlation of 50 % for all valid hourly observations over the period of record.

PA RH values were restricted to the range 30 %–70 % (any values above/below this were set to 30 % or 70 %, respectively) as these values are near the efflorescence and deliquescence points typical of fine particulate matter (Parsons et al. 2004, Davis et al. 2015). Corrections utilizing RH tended to overcorrect observations at these extreme RH values, and the observed bias becomes increasingly unpredictable (Figure SI.1). The RH data from the PA monitors were not corrected for the temperature bias resulting from the internal electronic heat produced near the sensor and direct solar radiation. Solar radiation impacts were too difficult to estimate given the variations in siting at each of the locations.

## 2.3 Correction Development

Multiple model forms were tested, including simple linear (Eq. 2), piecewise linear with up to three break points (Eq. 3), and polynomial models (Eq. 4); each with and without RH as an additional term. Truncating the RH data to 30 % - 70 % consistently improved the performance of RH-based models. A temperature term was also tested; however, its impact was found to be minimal and given the high correlation between temp and rh, rh was selected as the more important term. Piecewise models which were built starting from the second segment (fitting the mid-range data first) tended to perform better in the mid-range $PM_{2.5}$ concentrations than those built starting from the first segment (fitting the low-range data first). Models which use RH-based corrections to account for particle growth prior to correcting were also tested (Eq. 5; Crilley et al., 2018). The k value in this equation represents the hygroscopicity parameter, derived from κ–Köhler theory (Petters and Kreidenweis, 2007). Multiple k values were determined for the Eq. 5 model using regression across three 30 μg m$^{-3}$–40 μg m$^{-3}$ bins (Low: 0 μg m$^{-3}$–30 μg m$^{-3}$, Moderate: 30 μg m$^{-3}$–60 μg m$^{-3}$, High: 60 μg m$^{-3}$–100 μg m$^{-3}$) as well as anything greater than 100 μg m$^{-3}$ (Very High). An 'optimal' k value was then selected such that the moderate to high range of concentrations would be improved the most using our evaluation framework presented here. More complex models, such as neural network-based corrections, were not tested due to the difficulty of transferability and useability for real time data.

$$Corrected = a * PM_{2.5} + b \tag{2}$$

$$Corrected = PM_{2.5} < x: a * PM_{2.5} + b; \ x \leq PM_{2.5} < x_2: c * PM_{2.5} + d ..... \tag{3}$$

$$Corrected = a * PM_{2.5} + b * PM_{2.5}^2 + c \tag{4}$$

$$Corrected = \frac{PM_{2.5}}{1 + k/(100/RH - 1)} \tag{5}$$

Our correction development process was iterative and many model forms were tested. The current correction models commonly used are simple linear regressions (Kelly et al., 2019; Feenstra et al., 2019; LRAPA, 2019) or multiple regressions including an RH and/or a temperature term (Arden Dryer et al., 2019; Magi et al., 2019; Barkjohn et al., 2021). Segmented (or 'piecewise') linear regressions have also been developed (Malings et al., 2019) as well as non-linear RH-based adjustments (Chakrabarti et al., 2004; Crilley et al., 2018; Tryner et al., 2020). The models which will be presented here are those that continually performed the best as our evaluations changed and improved. No correction model will work perfectly in every location, so it is important to test which correction works best at a given location.

## 2.4 Correction Evaluation

The correction models were applied to the testing data set and a suite of statistical comparisons were made to evaluate how well the developed correction models perform. In addition, comparisons with corrections from other studies were made to compare with our results. Duvall et al. (2021) outline several key metrics to consider for small sensor performance: precision, bias and error, linearity, effects of RH and temperature, sensor drift, and accuracy at high concentrations. Evaluating precision is not

viable in this study given that many sites only had a single PA installed. We will evaluate bias, error, and linearity through our analysis, as well as the effects of RH. We found temperature impacts to be minimal for our dataset, especially when the impacts of RH were already considered. Sensor drift is outside of the scope of our study, and accuracy at high concentrations is less of a concern given our use of the AQHI+ scale and focusing on the moderate to high concentrations.

When evaluating correction models, it is important to use a suite of statistical evaluation techniques, as each technique will have its own interpretation. We feel that correlation, while commonly used, by itself is not a good comparative measure as simple linear models will not alter the correlation and more complex models tend to minimally change the correlation in our experience. Several comparative statistics (Eq. 6 – 8) are used commonly (Feenstra et al., 2019; Datta et al., 2020; Zamora et al., 2019; Tryner et al., 2020), including root mean square error (RMSE) and mean error/bias (ME/MB). However, these measures are not normalised making comparisons between sites difficult. Normalised statistical methods (Eq. 9 – 11) such as normalised RMSE (NRMSE) and mean fractional error/bias (MFE/MFB) allow for compatibility between sites/studies and reduce the locational context necessary to determine the significance of the reported improvement (Boylan and Russell 2006).

$$RMSE = \sqrt{\sum_{i=1}^{n}[(mod_i - obs_i)^2] \times \frac{1}{n}} \tag{6}$$

$$ME = \sum_{i=1}^{n}[|mod_i - obs_i|] \times \frac{1}{n} \tag{7}$$

$$MB = \sum_{i=1}^{n}[mod_i - obs_i] \times \frac{1}{n} \tag{8}$$

$$NRMSE = \frac{RMSE}{\sum_{i=1}^{n}[obs_i] \times \frac{1}{n}} \tag{9}$$

$$MFE = \sum_{i=1}^{n}\left[\frac{|mod_i - obs_i|}{(mod_i + obs_i) \times \frac{1}{2}}\right] \times \frac{1}{n} \tag{10}$$

$$MFB = \sum_{i=1}^{n}\left[\frac{mod_i - obs_i}{(mod_i + obs_i) \times \frac{1}{2}}\right] \times \frac{1}{n} \tag{11}$$

We used the AQHI+ scale to evaluate correction performance across a range of concentrations to make our evaluations relevant to health outcomes (eq. 1; BC Lung Association, 2019). This helps reduce biasing comparison statistics, both with the majority of low AQHI+ observations and the much less common, high concentrations that are much more important from a health perspective. As concentrations in the middle and high end of the AQHI+ scale have the greatest impact on human health and a person's decisions about changing their behaviour, performance at AQHI+ levels > 3 (> 30 µg m$^{-3}$) should be given greater consideration when selecting a correction model. Conversely, the observations of AQHI+ ≤ 3 (≤ 30 µg m$^{-3}$) represent most observations experienced in North American cities; better performance at these levels will ensure better day-to-day functionality of the correction but will have less impact from a health perspective.

## 3 Results and Discussions

The colocation site selection metric we used detected 86 potential colocation sites during this period in Canada and the United States. All sites had missing data, five sites had PA sensors with manually flagged invalid data, 65 had months where the temperature or RH were deemed too invariable to be outdoors, 67 had hours flagged invalid from differences between the A and B sensors within the PAs, and six sites had monitors with less than two months of valid data. Across all of these sites, 40.1% of the PM$_{2.5}$ observations were missing (either from the FEM or a PA), <0.0001% were manually flagged as invalid, 3% were

flagged as months where the PA was likely indoors, 2.3% were flagged by our PA A/B sensor comparison, and 1.3% were removed from PA monitors with less than two months of valid data.

Following the QA/QC process, 47 colocation sites were selected for developing and testing the corrections (Table 1; Figure 1). Of these 47 sites, 32 were selected for the training dataset (for developing the correction models) and the remaining 15 were used for the testing dataset. Training/testing sites were randomly selected then adjusted (again randomly) to ensure representativeness across geographic areas and concentration ranges. The mean start date was January 24, 2019, for the training sites and November 18, 2018, for the testing sites; all sites had data until the end of the year 2020. The number of days with valid colocation data at each site ranged from 212.5 to 1258.5 (nearly 3.5 years); with an interquartile range (IQR) of 379 days to 723 days. For the training sites the hourly valid data capture ranged from 42.4 % to 98.9 %, with an IQR of 77.1 % to 92.5 %. The testing sites were similar, with valid data captures ranging from 66.7 % to 98.6 % and an IQR of 80.4 % to 88.1 %. The mean PA PM$_{2.5}$ concentration was larger than that from the FEM monitor for all but five sites. Across all sites, 0.75% of the RH observations were missing and replaced with a value of 50%, 17% of the observations had an RH less than 30% (replaced with a value of 30%), and 10.5% had an RH greater than 70% (replaced with a value of 70%).

**Table 1. Selected PurpleAir (PA)/FEM colocation sites (Tr. = Training, Te. = Testing). Pearson's Correlation (*r*) between valid PA/FEM PM2.5 observations provided for each site. IDs corresponding to the map in Figure 1 are provided. Detection methods are provided for the site where this information was available (Beta = Beta Attenuation, FDMS = FDMS Gravimetric, Spec. = Broadband Spectroscopy).**

| ID | FEM Site Name | Set | Detector | # PAs | Days Valid | r | ID | FEM Site Name | Set | Detector | # PAs | Days Valid | r |
|---|---|---|---|---|---|---|---|---|---|---|---|---|---|
| 1 | Abbotsford Airport | Tr. | Beta | 3 | 844.6 (98.9%) | 0.91 | 25 | Redwater | Tr. | Spec. | 1 | 662.9 (86.2%) | 0.97 |
| 2 | Blair Street | Tr. | FDMS | 1 | 383 (88.2%) | 0.61 | 26 | Riverside - Rubidoux | Tr. | Beta | 9 | 1169.8 (95.3%) | 0.86 |
| 3 | Boston - Von Hillern | Tr. | Beta | 2 | 266.9 (94.4%) | 0.70 | 27 | San Diego- Sherman Elem. | Tr. | N/A | 3 | 288.2 (85.4%) | 0.87 |
| 4 | Carpenter | Tr. | Beta | 2 | 1127.3 (91.9%) | 0.63 | 28 | Santa Barbara | Tr. | Beta | 1 | 588.8 (90.2%) | 0.81 |
| 5 | Ctclusi Radar Hill - OR | Tr. | N/A | 1 | 293.5 (63.7%) | 0.87 | 29 | Santa Maria - Broadway | Tr. | Beta | 1 | 614 (88.1%) | 0.83 |
| 6 | Darrington – Fir St | Tr. | Beta | 1 | 398.7 (90.6%) | 0.95 | 30 | Tacoma - Alexander Ave | Tr. | N/A | 1 | 756.8 (98.8%) | 0.91 |
| 7 | Downtown Sacramento | Tr. | Beta | 29 | 717.4 (89.2%) | 0.97 | 31 | Thousand Oaks | Tr. | Beta | 1 | 562.5 (92.6%) | 0.82 |
| 8 | Edmonton Woodcroft | Tr. | FDMS | 1 | 374 (95.4%) | 0.75 | 32 | Vancouver - NE 84th Ave | Tr. | Beta | 1 | 428.8 (54.7%) | 0.92 |
| 9 | Rio Mesa School #2 | Tr. | Beta | 1 | 694.1 (90.9%) | 0.73 | 33 | Arendtsville | Te. | Beta | 2 | 404.2 (86.3%) | 0.90 |
| 10 | Fresno - Garland | Tr. | Beta | 2 | 758.8 (80%) | 0.92 | 34 | Atascadero2 | Te. | Beta | 1 | 916 (82.4%) | 0.89 |
| 11 | Eastgate | Tr. | Spec. | 1 | 258.9 (59.9%) | 0.89 | 35 | Bee Ridge | Te. | Spec. | 1 | 311.8 (60.9%) | 0.80 |
| 12 | Indpls - Washington Park | Tr. | Spec. | 1 | 728.3 (71.4%) | 0.75 | 36 | Burnaby South | Te. | Beta | 1 | 411.5 (91.4%) | 0.93 |
| 13 | Lake Sugema | Tr. | N/A | 2 | 448.5 (96%) | 0.75 | 37 | Eugene - Highway 99 | Te. | Beta | 1 | 838.5 (97%) | 0.94 |
| 14 | Lancaster Dw | Tr. | Beta | 1 | 355.2 (77.7%) | 0.90 | 38 | Grass Valley | Te. | Beta | 1 | 370.5 (80.8%) | 0.97 |
| 15 | Lee Vining | Tr. | Spec. | 1 | 416.8 (90.2%) | 0.99 | 39 | Greenwood | Te. | Spec. | 1 | 473.2 (76%) | 0.91 |
| 16 | Math & Science Ctr | Tr. | Spec. | 1 | 583.8 (84.4%) | 0.90 | 40 | Hillsboro - Hare Field | Te. | N/A | 1 | 400.3 (92%) | 0.99 |
| 17 | Mcmillan Reservoir | Tr. | Beta | 1 | 212.5 (62%) | 0.75 | 41 | Iowa City, Hoover School | Te. | Spec. | 2 | 715.9 (98.3%) | 0.90 |
| 18 | Oakridge - Willamette | Tr. | Beta | 3 | 439.5 (99.4%) | 0.98 | 42 | Keene | Te. | Beta | 2 | 936.3 (89.5%) | 0.82 |
| 19 | Ojai - Ojai Ave. | Tr. | Beta | 1 | 276.8 (90.5%) | 0.85 | 43 | LLCHD BAM | Te. | N/A | 1 | 654.2 (66.4%) | 0.70 |
| 20 | Pendleton - Mckay Creek | Tr. | N/A | 1 | 355.7 (55%) | 0.97 | 44 | Ncore | Te. | N/A | 2 | 374.6 (67.2%) | 0.89 |
| 21 | Piru - Pacific | Tr. | Beta | 1 | 504.6 (80.8%) | 0.79 | 45 | White Mt Research Cntr. | Te. | N/A | 1 | 664.9 (88.8%) | 0.95 |

| 22 | Portland - Se Lafayette | Tr. | N/A | 1 | 775.3 (96.6%) | 0.96 | 46 | PRG Plaza 400 | Te. | Beta | 3 | 1258.5 (97.8%) | 0.94 |
| 23 | Public Health | Tr. | Spec. | 2 | 872.5 (94.5%) | 0.84 | 47 | Simi Valley - Cochran St. | Te. | Beta | 1 | 675.9 (87.4%) | 0.87 |
| 24 | Red Bluff - Walnut Office | Tr. | Beta | 1 | 602.8 (73.9%) | 0.92 | | AVERAGE | - | | 2.1 | 578 (84.5%) | 0.87 |

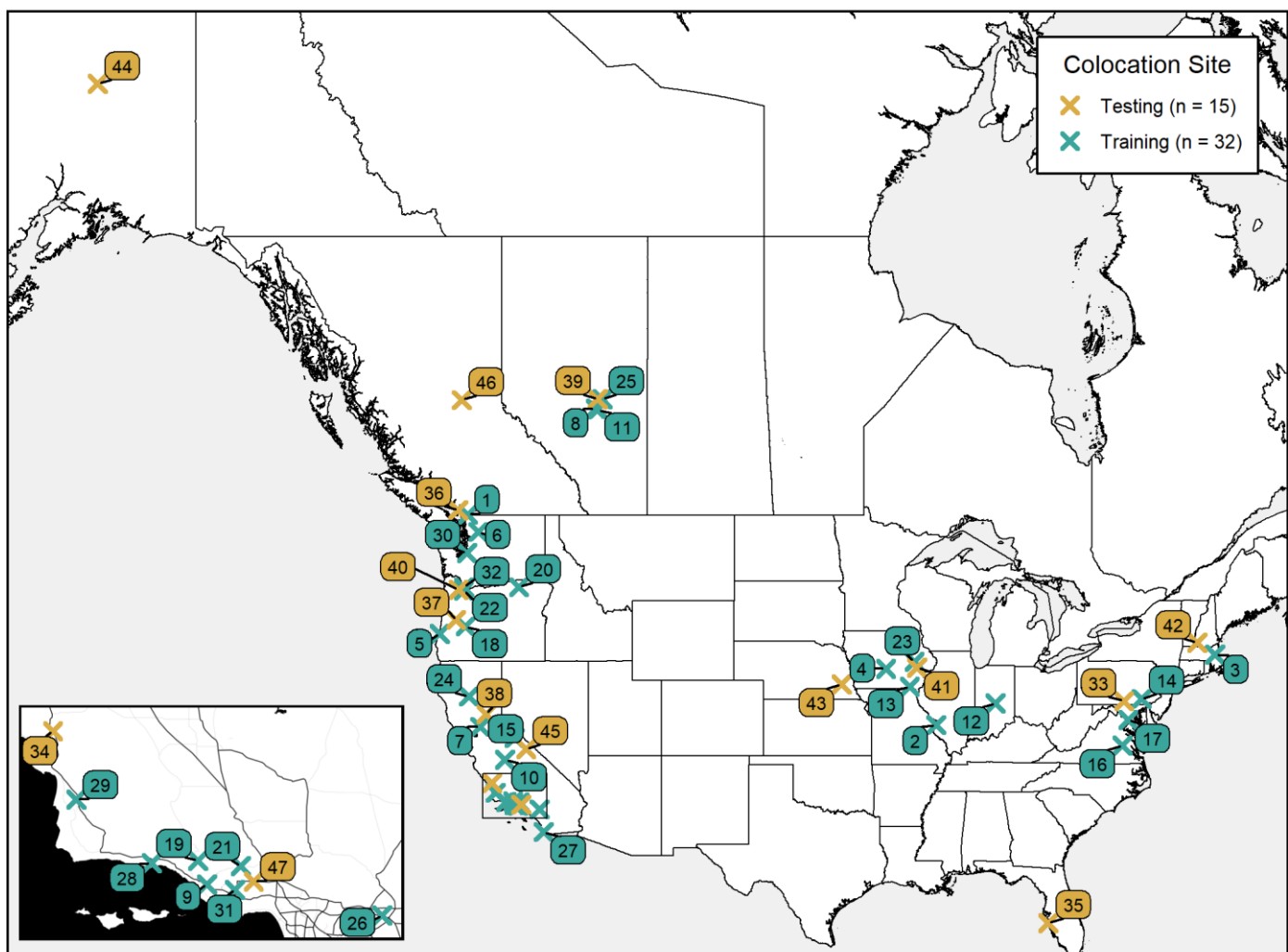

**Figure 1: Locations of selected PurpleAir/FEM colocation sites. Of the 47 sites, 32 were used for training the correction models and the remaining 15 were used for testing. Inset map tiles by Stamen Design, under CC BY 3.0. Data by OpenStreetMap, under ODbL.**

Hourly concentrations of PM$_{2.5}$ ranged between $0 - 837$ µg m$^{-3}$ and $0 - 986$ µg m$^{-3}$ across all sites during this period for PA and FEM monitors, respectively (Figure 2). PA monitors at most sites tended to be within a factor of 2 of FEM, typically biased higher. For most sites this bias appears to be linear as concentrations increase. PA PM$_{2.5}$ concentrations across all sites were

245 categorised as "Low AQHI+" ($0 - 30$ µg m$^{-3}$) for 91.1% of observations, "Moderate AQHI+" ($30 - 60$ µg m$^{-3}$) for 7.7%, "High AQHI+" ($60 - 100$ µg m$^{-3}$) for 0.7% and "Very High AQHI+" ($100+$ µg m$^{-3}$) for 0.6% of observations. In the same order for the FEM monitors at all sites: 97.5%, 1.9%, 0.3% and 0.4% of observations were in the four AQHI+ categories.

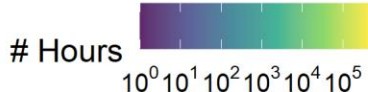

Figure 2. Hourly observation counts of QA/QC'ed FEM vs. PA PM$_{2.5}$ concentrations across all colocation sites in this study. The dashed lines are the 0.5:1, 1:1, and 2:1 lines. The solid lines are individual site General Additive Models (GAMs).

The mean averages of the PA PM$_{2.5}$ concentrations at each site tended to be higher than their FEM counterparts in both data sets except at the lowest humidity levels (Figure 3). For the training data, 23% of the hourly observations were classified as "Dry" (RH ≤ 33%), 56% were classified as "Moderate" (30% < RH < 70%), and 21% were classified as "Humid" (RH ≥ 70%). The testing data were similar with 25%, 55%, and 19% of the hours classified as dry/moderate/humid, respectively. PA concentrations tended to be biased increasingly higher as humidity increased; this was not the same case for the FEM monitors which typically have their intake stream heated or dried to lower the RH impacts. Both the FEM and PA monitors had similar ranges of site median concentrations between the testing and training data sets; however, many of the testing data sites tended to have lower concentrations on average than the training sites.

The FEM monitors in this study utilized one of three main $PM_{2.5}$ detection methods (Figure 4): beta attenuation (26 locations), broadband spectroscopy (9 locations) and FDMS gravimetric (2 locations). There were ten locations with unknown monitor types, all in the USA. It is most likely that these are beta attenuation monitors based on the high ratio of these detectors in the USA versus others and the comparability between site mean AQHI+ error shown in Figure 3; however, some may be broadband spectrometers or gravimetric monitors. There were no significant differences between the mean of the site mean PA AQHI+ errors for each of the FEM sensor types (Kruskal–Wallis $p > 0.05$) when ignoring the FEM observations with an AQHI+ of 1, as these make up the bulk of the observations and are unimportant from a health and management perspective.

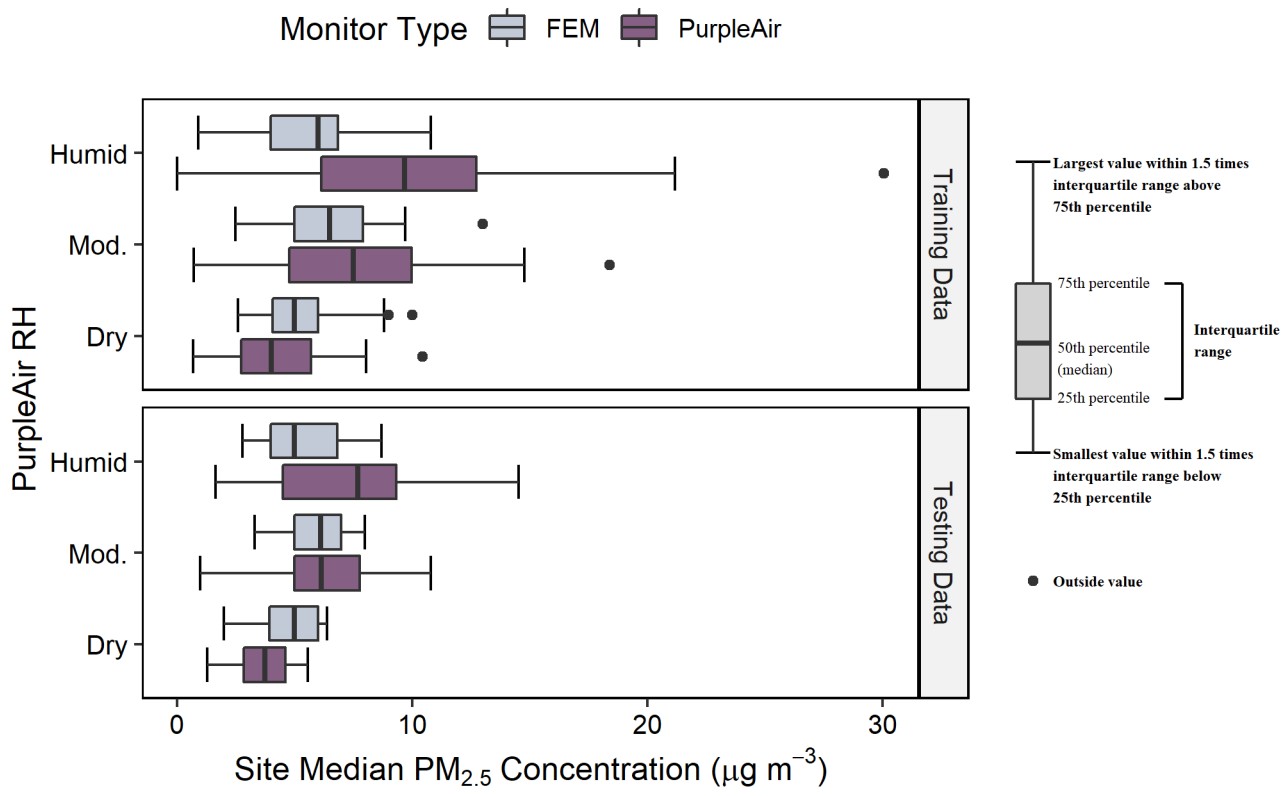

**Figure 3. Distributions of the Federal Equivalent Method (FEM) and PurpleAir training/testing sites median PM2.5 concentrations (µg m⁻³) at dry (0 %–33 %), moderate (34 %–66 %) and humid (67 %–100 %) relative humidity (RH) groupings. The boxplot format is indicated on the right.**

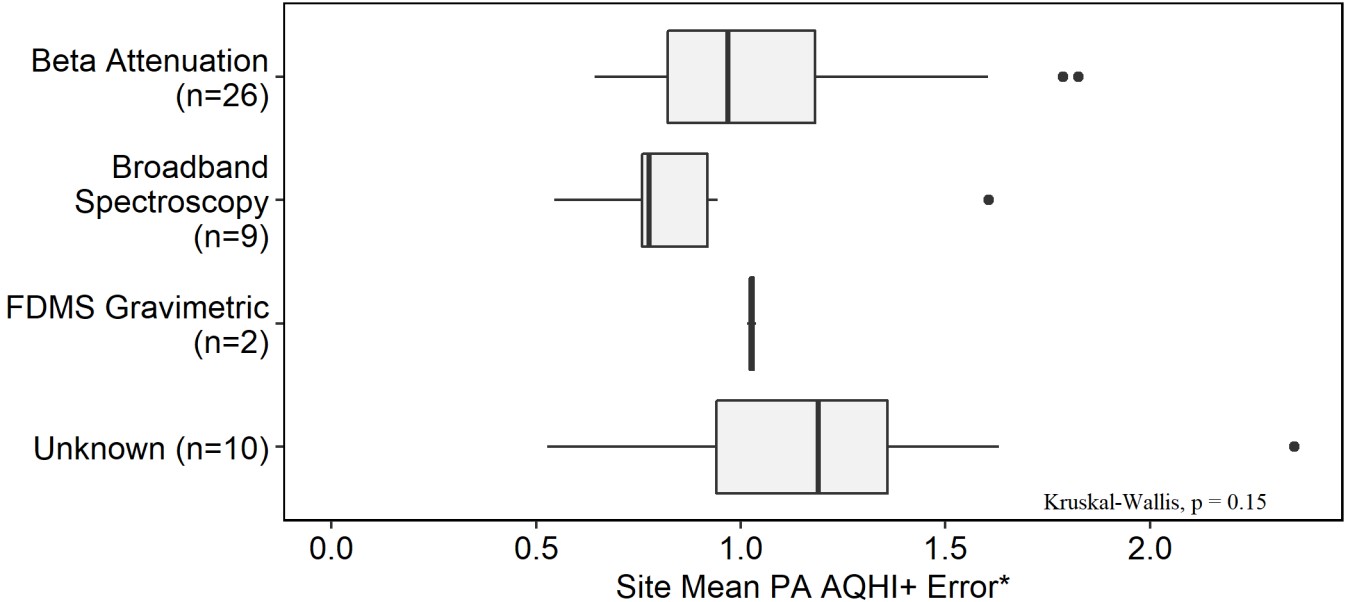

Figure 4. Distributions of the site mean PurpleAir AQHI+ error for each of the known FEM detector types. Hours where the Federal Equivalent Method (FEM) AQHI+ was equal to 1 were removed as these make up the bulk of the observations and are unimportant from a health and management perspective. The Kruskal–Wallis global p-value (>0.05) is provided in the bottom right, indicating no significant differences between the group means.

## 3.1 Correction Development

Eight correction models were selected for evaluation (Table 2) including four developed in this study from regressing the training data (Models 1–4) as well as four others available from the literature (Models 5–8). Nearly all these correction models utilize the 'PM$_{2.5}$ CF 1' column provided by PA, except Model 6 and Model 8 which use 'PM$_{2.5}$ CF ATM'. Model 1 is a multiple linear model including a RH term. Model 2 uses a RH growth adjustment (k=0.24) to reduce the PM$_{2.5}$ concentration as RH increases (see Eq. 5). Model 3 is a second-degree polynomial with a RH term included. Model 4 is a 3-breakpoint piecewise model with breakpoints selected to visually fit the data best over multiple iterations. The last four equations are provided from studies by other parties and consist of two simple linear models (Models 5 and 6), a multiple linear model including RH (Model 7) and a multiple linear model including both RH and temperature (Model 8).

Table 2. PurpleAir correction models selected for evaluation. The 'Min' column indicates the minimum corrected value at a RH of 70% (*and a temperature of 20 ºC for Model 8).

| Correction | Form | Source | Min | Formula | |
|---|---|---|---|---|---|
| No Model | - | - | 0 | pm25_cf1 | |
| Model 1 | Linear (+RH) | - | -2 | 0.708 * pm25_cf1 - 0.115 * rh + 5.78 | |
| Model 2 | RH Growth | - | 0 | pm25cf1 / (1 + 0.24 / (100 / rh - 1)) | |
| Model 3 | Polynomial (+RH) | - | 0.3 | 0.53 * pm25_cf1 + 0.000952 * pm25_cf1^2 - 0.0914 * rh + 6.3 | |
| Model 4 | Piecewise @ 2.5/40/300 | - | 1.9 | pm25_cf1 ≤ 2.5: <br> 2.5 < pm25_cf1 ≤ 40: <br> 40 < pm25_cf1 ≤ 300: <br> pm25_cf1 > 300: | 0.92 * pm25_cf1 + 1.86 <br> 0.42 * pm25_cf1 + 3.1 <br> 0.87 * pm25_cf1 - 14.8 <br> 1.16 * pm25_cf1 - 100.6 |
| Model 5 | Linear | A | 2.6 | 0.778 * pm25_cf1 + 2.65 | |
| Model 6 | Linear | B | -0.7 | 0.50 * pm25_atm - 0.66 | |
| Model 7 | Linear (+RH) | C | -0.2 | 0.534 * pm25_cf1 - 0.0844 * rh + 5.71 | |
| Model 8 | Linear (+RH +T) | D | 1.5* | (pm25_atm + 3.04 + 0.07 * temp - 0.02 * rh) / 1.55 | |

A. Kelly et al. (2017) B. LRAPA (2019) C. Barkjohn et al. (2021) D. Ardon-Dryer et al. (2019)

The corrected data from the models typically had a minimum value of at or just below 0 µg m$^{-3}$, except for Models 4, 5, and 8 which had a higher minimum value (around 2 µg m$^{-3}$). Models 1, 6 and 7 produced negative values during periods of low

concentrations and high humidity (which need to be removed or replaced with 0 after correcting). A constant temperature of 20 ºC was assumed here for Model 8 to calculate the minimum corrected value.

## 3.2 Correction Evaluation

Raw PA observations were biased positively at FEM AQHI+ levels between 1 and 9, peaking at an AQHI+ of 4 to 7, as shown in Figure 5. An AQHI+ bias at or near zero is preferred, especially at the higher FEM AQHI+ levels most impactful to human health. Models 2, 5, and 7 minimised this bias the most on average. Model 2 was biased slightly high at AQHI+ of 6 or lower, and slightly high onwards with a minimum at 10 AQHI+. Models 5 and 7 performed similarly; however, they were biased slightly low throughout, except at an AQHI+ of 1, and had the worst performance at 8 and 10 AQHI+. Models 1 and 8 were the next best; however, they were both increasingly negatively biased at higher FEM AQHI+ levels. Model 6 had relatively large negative biases in the 3–10+ AQHI+ levels. A detailed breakdown for each testing site can be found in the SI (Figure SI.4). Further comparisons were only made on Models 1, 2, 5, and 7 as they showed the best performance here. Comparisons for the remaining models can be found in the SI.

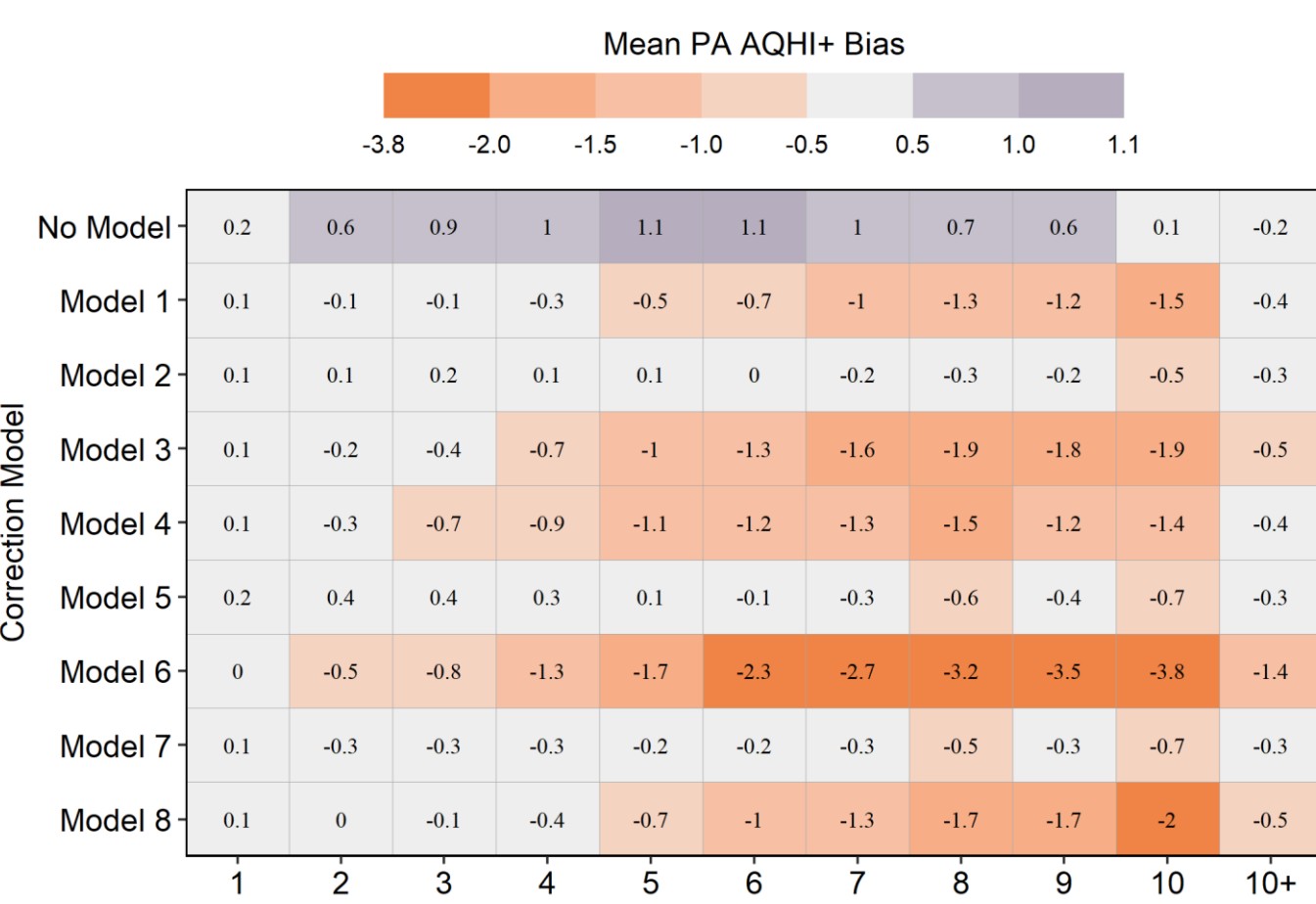

**Figure 5. Mean PurpleAir AQHI+ bias for each correction model (including the raw data) at Federal Equivalent Method (FEM) AQHI+ levels. A value at or near 0 is preferred, especially for higher AQHI+.**

The NRMSE, NMB, MFE and MFB for the PA monitors were worse at lower concentrations and improved as concentrations increased (Figure 6). We saw similar performance between models; all of which improved upon the raw PA observations at most concentrations. Model 1 worsened the MFE and bias measurements in the moderate to high range, Model 5 performed poorly

below an AQHI+ of 5, and Models 2 and 7 had the best performance outside of the low concentration range. Model 7 was best for the very low observations (AQHI+ of 1); however, Model 2 tended to perform better starting at 2–3 AQHI+.

Goal and acceptable levels for MFE and MFB are suggested in Boylan and Russell (2006). Raw PA data meets the goal level of 50 % for all but the lowest AQHI+ for MFE where it still meets the acceptable level of 75 %. Only Models 7 and 8 bring these lowest concentrations into/near the goal level. Both uncorrected and corrected observations were within the goal range for MFB (±30 %). We assumed goal and acceptable levels for NRMSE of 50 % and 70 %, respectively, to align with the levels defined for MFE. Using these standards, the uncorrected PurpleAir data is unacceptable at AQHI+ equal to 1 and gets increasingly better until it reaches the goal level in the high concentrations. Each correction model brings the data into the goal level except at AQHI+ equal to 1 where it is only acceptable. A goal level of ±30 % was assumed for the NMB like that for the MFB and the level defined for mean bias (MB) defined in Chang and Hanna (2004). Only the uncorrected data for AQHI+ values between 2–3 exceeded this goal level across our sites.

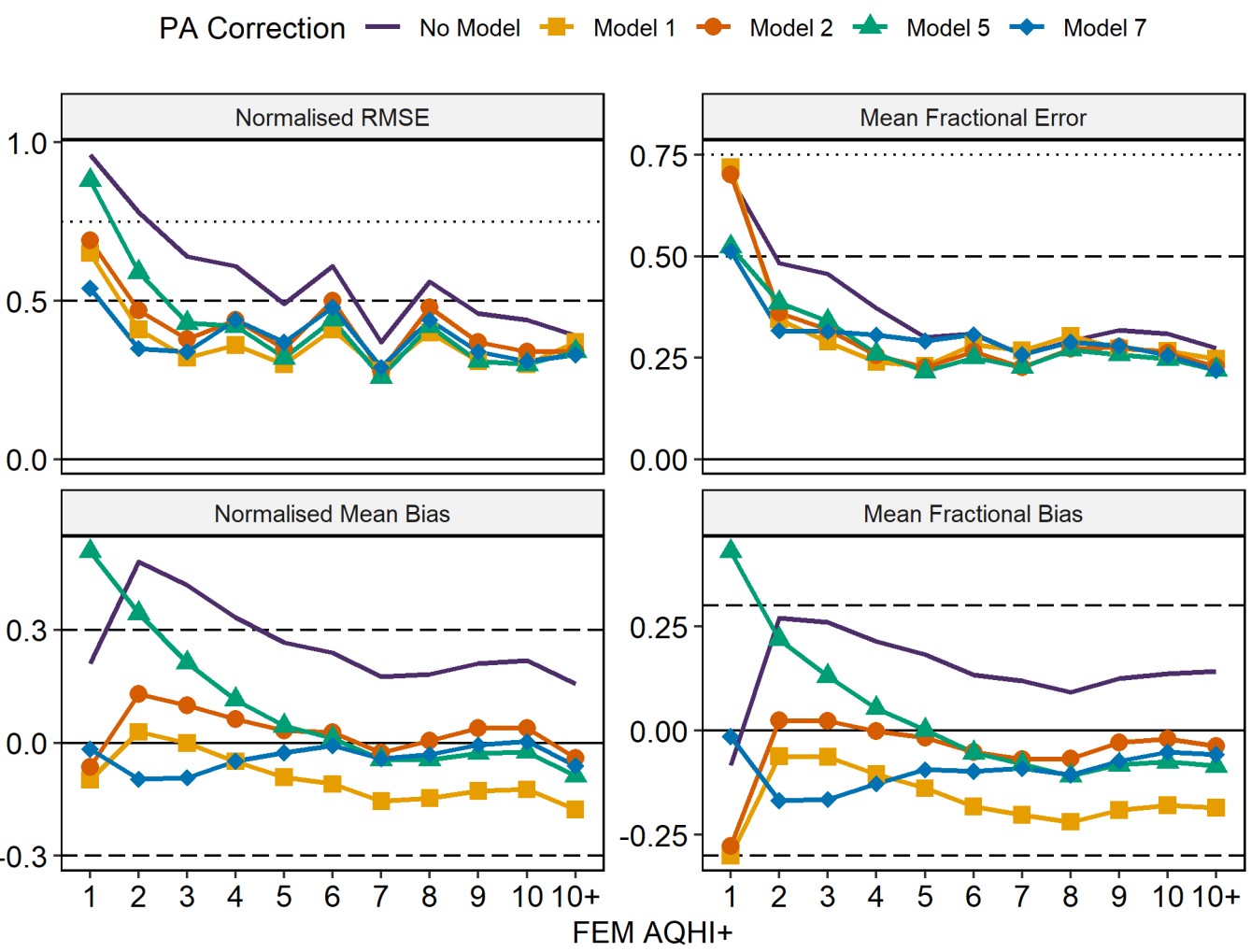

**Figure 6. Comparison statistics across Federal Equivalent Method (FEM) AQHI+ levels for select correction models. Goal and acceptable levels are displayed where possible for RMSE (0.75 & 0.5), MFE (0.75 & 0.5), NMB (±0.3 & ±0.6) and MFB (±0.3 & ±0.6).**

These statistical comparisons were also made on coarser AQHI+ groupings (Table 3). A ranking score was calculated for each model using the mean average of the ranks (from 1 to 5) for each statistic within an AQHI+ group. A lower score indicates better relative performance within that AQHI+ range. Models 2, 5 and 7 performed the best in the high and very high AQHI+

categories. However, the models did not perform as consistently well in the low and moderate ranges. Models 2 and 7 tended to be the best for the low range, while Models 1, 2 and 5 were better for the moderate range.

The error/bias in AQHI+ from the uncorrected PA monitors was greatest in the moderate to high range (Figure 7). The selected corrections produced similar improvements to each other, all improving upon the PA's ability to correctly determine the hourly AQHI+ except for at the most extreme concentrations (> 100 µg m$^{-3}$). For AQHI+ bias, Model 1 improved upon Model 7, which

has the same form, except at very high FEM concentrations. This was not the case for the mean AQHI+ bias; however, as Model 1 becomes increasingly negatively biased as FEM concentrations increase. Model 2 was the best overall performer for both AQHI+ error and bias, being marginally outperformed only in the low concentration range. This was followed by Models 5 and 7 which performed well except for the mean AQHI+ error in the moderate concentration range for Model 7.

**Table 3. PurpleAir (PA) normalised mean bias (NMB), normalised root mean square error (NRMSE), mean fractional error (MFE)**
**and mean fractional bias (MFB) at low/moderate/high/very high Federal Equivalent Method (FEM) AQHI+ levels for each PA correction model. A crude score is calculated by averaging each statistics integer rank (from 1 to 5) for the models within that AQHI+ group. The top performing models are highlighted.**

| FEM AQHI+ | Model | NMB | NRMSE | MFE | MFB | Score |
|---|---|---|---|---|---|---|
| | No Model | 0.33 | 0.96 | 0.65 | **-0.01** | 3.8 |
| | Model 1 | -0.04 | **0.56** | 0.64 | -0.25 | 3.0 |
| Low [1-3] | Model 2 | **0.02** | 0.62 | 0.63 | -0.22 | 2.5 |
| | Model 5 | 0.42 | 0.77 | 0.49 | 0.39 | 4.0 |
| | Model 7 | -0.05 | 0.49 | **0.47** | -0.04 | **1.8** |
| | No Model | 0.30 | 0.59 | 0.35 | 0.20 | 5.0 |
| | Model 1 | -0.07 | **0.37** | **0.24** | -0.12 | **2.3** |
| Moderate [4-6] | Model 2 | 0.05 | 0.44 | 0.25 | **-0.01** | 2.4 |
| | Model 5 | 0.08 | 0.40 | 0.25 | 0.03 | 2.5 |
| | Model 7 | **-0.03** | 0.44 | 0.30 | -0.11 | 2.9 |
| | No Model | 0.19 | 0.47 | 0.28 | 0.11 | 4.8 |
| | Model 1 | -0.15 | **0.33** | 0.28 | -0.20 | 3.6 |
| High [7-9] | Model 2 | **0.00** | 0.38 | **0.25** | **-0.06** | 2.0 |
| | Model 5 | -0.04 | **0.33** | **0.25** | -0.09 | **1.9** |
| | Model 7 | -0.03 | 0.36 | 0.27 | -0.09 | 2.8 |
| | No Model | 0.16 | 0.39 | 0.28 | 0.14 | 4.5 |
| | Model 1 | -0.17 | 0.38 | 0.25 | -0.18 | 4.5 |
| Very High [10+] | Model 2 | **-0.04** | **0.34** | 0.23 | **-0.04** | **1.8** |
| | Model 5 | -0.08 | **0.34** | **0.22** | -0.08 | 2.3 |
| | Model 7 | -0.06 | **0.34** | **0.22** | -0.06 | 2.0 |

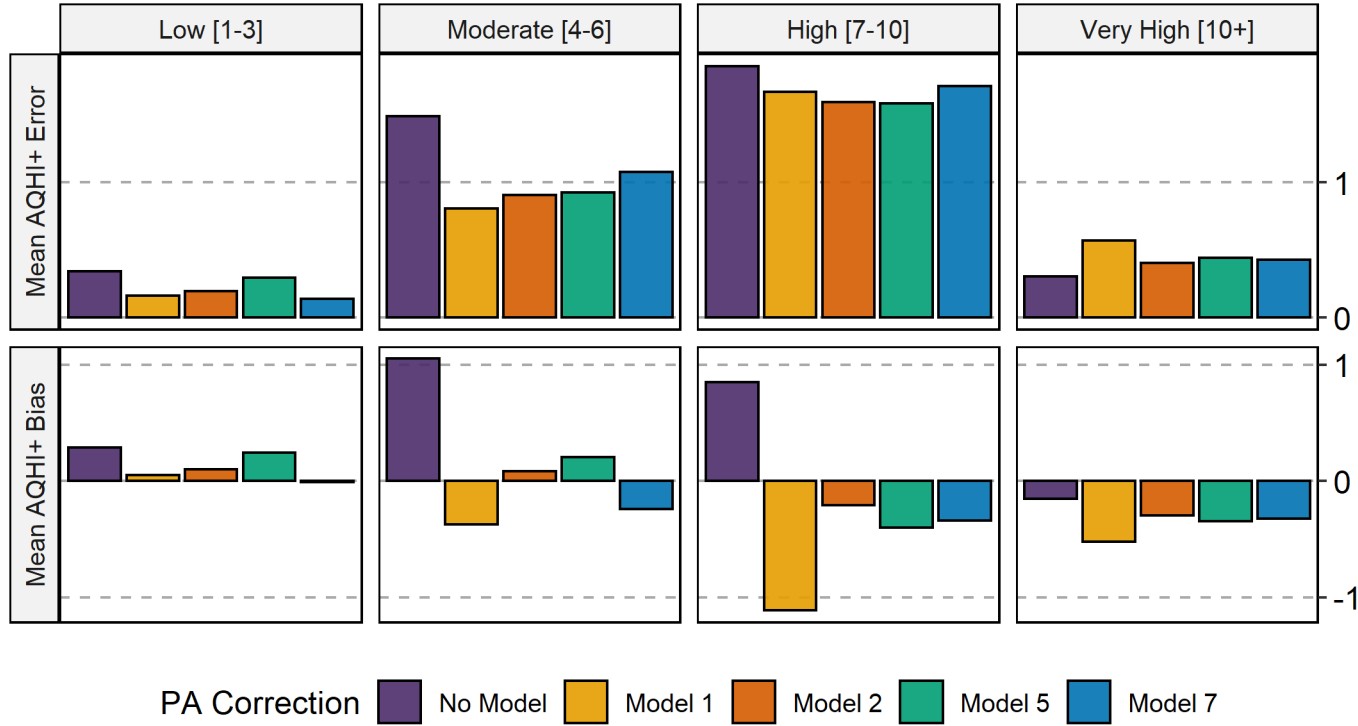

 **Figure 7. Mean PurpleAir (PA) AQHI+ error and bias for low/moderate/high/very high Federal Equivalent Method (FEM) AQHI+ levels for selected correction models.**

Model 2 produces a mean AQHI+ bias and error consistently closer to 0 than the other correction models (or no model at all) across the range of observed concentrations. Only at the lowest concentrations do the other models marginally outperform Model 2 across our testing sites for most of the comparisons made.

**4. Conclusions**

The concentrations of PM$_{2.5}$ reported from the PA monitors were biased high compared to the FEM monitors at most colocation sites, especially for the lower concentration range. This bias is attributed to both the method of detection as well as hygroscopic growth from increased humidity. No significant differences were detected between FEM sensor methods, which are rigorously tested and evaluated to ensure this consistency. Based on our colocation method there is a risk that some sites are not true colocation sites, however, given the proximity of the sensors and good correlation (>60 %) we decided the potential impact would be minimal. The Canadian AQHI+ system was useful as a framework for evaluating correction models across a range of concentrations, as infrequent high values or numerous low values can skew performance statistics when evaluating the full range at once.

The selected corrections discussed here improve the performance of PA similarly, however, Model 2 (our "RH Growth" model) had consistently better performance, especially at moderate to high concentrations that are important to health. This was followed closely by Model 7, the multiple (RH) linear regression from Barkjohn et al. (2021) and Model 5, the simple linear regression from Kelly et al. (2017). Model 1, our multiple linear regression with the same form as Model 7, performed well in the low to moderate range, however it did not perform as well at higher concentrations. It should be noted that the average performance across the testing sites and over time was evaluated here; performance at colocation sites and across time was not

the same (see SI Figure 4). In addition, while our correction model focuses on correcting the impacts of humidity, other characteristics like refractive index and particle shape, density, and size distribution may account for differences in PM$_{2.5}$ estimates.

Future studies should focus on developing and evaluating correction models as more data become available from the PA network, as well as installing and documenting additional colocation sites. Specifically, the k-value utilized in our Model 2

should be adjusted and re-evaluated as more collocation data become available. Valid collocation data are imperative for both developing and evaluating small sensor performance before and after applying a correction formula. These corrections should be evaluated in comparison with the others available to ensure comparable or improved performance.

We recommend testing the performance of several models at specific sites of interest and selecting the best performing model for a given site (Figure 4 in the SI provides a breakdown for the testing sites and models evaluated here). Models 1, 2, 5 and 7

presented here are good starting points. As more colocation data become available, seasonal and area specific correction models should be examined. Performance in the moderate to high concentration range should be focused on as these are the most important from a health perspective; the low concentrations are less important while also being the most observed levels in the US and Canada. Correlation is useful for evaluating overall sensor performance at a site, but not as useful for evaluating and comparing correction performance. Normalised methods such as NRMSE, MFB or MFE are good measures, but we recommend

evaluations across a range of PM$_{2.5}$ concentrations, such as using the AQHI+ framework as presented here. If one intends to develop a site-specific correction model, we recommend using the same form as our Model 2 while adjusting the k value. For scenarios where testing models on individual locations is not an option, such as applying a correction in an area without a nearby PA-FEM colocation site, we recommend using our Model 2.

## 5. Author Contribution

B. Nilson developed the analysis code and drafted the initial manuscript with contributions from all authors. P. L. Jackson, C. Schiller, and M. T. Parsons provided regular guidance and feedback throughout the project.

## 6. Code/Data Availability

Code/data from this study are available upon request to the authors.

## 7. Competing interests

The authors declare that they have no conflict of interest.

## 8. Acknowledgements

Data sourced from the PurpleAir ThingSpeak database (PurpleAir) and the US AirNow database (FEM). Funding and Support from ECCC and from the Natural Sciences and Engineering Research Council of Canada (NSERC) through a Discovery Grant (RGPIN2017-05784) to Peter Jackson. Thanks to the anonymous referees for their positive contributions to the paper during the
390 review process.

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
