# Peer review of "Development and Evaluation of Correction Models for a Low-Cost Fine Particulate Matter Monitor"

_Atmospheric Measurement Techniques, 2021_

## Author Response (AR1)

**Referee #1 Response Letter**

Thank you for the time you put into reviewing our manuscript and the very useful and helpful feedback which has led to improvements in the paper. Please see our following responses and proposed alterations which we believe will resolve your individual comments.

**Comment 1:**

Were the authors trying to develop simple models to correct ambient PM2.5 concentrations reported by PurpleAir monitors across a range of locations and seasons in Canada and the United States? If so, how does this work build upon and differ from that of Barkjohn et al. [DOI: 10.5194/amt-14-4617-2021]?

Response 1:

See response #2

This work differs from and builds upon that of Barkjohn et al. since we include comparisons of the model from Barkjohn plus several other sources, our selected model form (Model 2) is different, we evaluate on an hourly time scale (not 24-hour averages like Barkjohn et al.), we include Canadian sites, and we use the AQHI+ system.

**Comment 2:**

It sounds like the authors might have been more interested in developing correction models that would help PurpleAir monitors predict "high" and "very high" AQHI+ levels correctly and expected wildfire smoke to be the most common cause of high or very high AQHI+ levels in Canada and the US. If so, the manuscript would benefit from discussions of (a) the existing literature describing the PurpleAir response to wildfire smoke [for example, Delp and Singer, DOI: 10.3390/s20133683 and Holder et al., DOI: 10.3390/s20174796] and (b) how the PurpleAir response to smoke is affected by the physics governing the PMS5003 sensor operation. Results from multiple field studies indicated that PMS5003 sensors overestimate smoke concentrations, and recent work by Ouimette et al. [DOI: 10.5194/amt-2021-170] indicated that this overestimation might be due to the small size of the particles produced during combustion. It is unlikely that the PMS5003 overestimates PM concentrations associated with wildfire smoke because of hygroscopic growth alone, so I'm a bit concerned by the authors' conclusion that Model 2, which was designed to account primarily for RH effects, is best for correcting PurpleAir data reported when PM2.5 concentrations are "high" or "very high". I guess the k-value that was fit empirically helps account for some of the other factors.

Response 2:

We never stated that the "high" and "very high" AQHI+ levels should be focused on for correction performance. We state multiple times (see lines 17, 99, 154, 301, 317) that the "moderate" to "high" levels should be focused on, given that health messaging does not change

until >3 AQHI+ and when AQHI+ is greater than 10 the messaging is the same (air quality is poor). We say that models 2 and 7 consistently perform better in high and very high on line 269, but they also perform well below that (just some models perform as well or slightly better). As well as on line 275 we say model 1 performs worse than model 7 in the very high range, but we don't discuss model 1 much otherwise. Nowhere else do we mention "very high" concentrations aside from when describing AQHI+ categories.

In our revised document we made edits to our introduction and added the following sentence to more clearly state our study objectives.

"It is our objective to create a correction model for general application across multiple sensors/locations, however, a more specialised correction is recommended where nearby colocation data are available."

**Comment 3:**

Regardless of the authors' objectives, it would be great to see more discussion of the physics that govern PMS5003 sensor operation and parameters that can affect the accuracy of PM2.5 concentrations reported by these sensors.

Response 3:

We added more information on how the nephelometers operate to our paragraph in the introduction already discussing the PurpleAir monitors and potential errors.

"[...]; concentrations are derived by correlating this scattering amplitude with a mass-based monitor (Hagan and Kroll, 2020)."

**Comment 4:**

Considering that the relationship between the ambient PM2.5 concentration and the light scattering signal received by the PMS5003 depends on particle size distribution, shape, refractive index, and density, and that these parameters can vary with location, season, and day-to-day weather, why do the authors conclude that a simple correction such as Model 2 or Model 7 is likely to produce acceptable results across a wide range of locations and seasons? Are there data on ambient aerosol properties in Canada and the US that the authors can cite to support this argument?

Response 4:

We acknowledge that simple corrections will never be perfect in all circumstances given the limitations imposed by the hardware and the wide range of particle properties that can exist and that a nephelometer cannot differentiate; we state on line 322 that where possible a site specific correction model should be developed, this is meant to be a general use model that performs better on average at most locations. On line 317 we state that in the future a seasonal/location specific model should be developed.

**Comment 5:**

I appreciate simple correction approaches because they contribute to the goal of making PM2.5 monitors and monitoring data accessible to a wide range of stakeholders, but I also think readers will have reasons to be skeptical of these correction models, and the authors should acknowledge and discuss those concerns.

Response 5:

We added the following sentence warning about the concerns behind using these models generally.

"Caution is advised when utilizing generalised models such as these, as they will not provide the same degree of improvement at all locations given differences in aerosol properties that a nephelometer cannot detect or differentiate."

**Comment 6:**

The authors argue that it's most important for PurpleAir monitors to predict "high" and "very high" AQHI+ values correctly, but I'm not sure I agree. When PM2.5 concentrations are high or very high, it's usually clear to the public that air quality poses a health hazard. I think it might be more important for the PurpleAir to predict AQHI+ values between 2 and 6 correctly, so that they can alert the public to pollution levels that are elevated above the baseline AQHI+ of 1 but that individuals might not readily perceive as hazardous using their senses. This comment is just for the authors' consideration.

Response 6:

See response #2

**Comment 7:**

1. Lines 72-73: "PM2.5 concentration is reported by the sensors using two different proprietary correction factors (PM2.5 CF 1 and PM2.5 CF ATM) which convert the estimated particle count in size bins into the reported concentrations." This statement is speculative at best and most likely incorrect. I suggest the authors delete it. As the authors describe on lines 58-62, the PMS5003 functions as a nephelometer and not an optical particle counter. Ouimette et al. [DOI: 10.5194/amt-2021-170] have shown that the sensor output labeled "number of particles with diameter beyond 0.3 μm in 0.1 L of air" is a measure of the amount of light scattered to the detector by particles passing through the sensor. The relationship between the amount of light scattered to the detector and the PM2.5 concentration reported by the sensor is unknown, but multiple published datasets indicate that the particle count distribution output by the PMS5003 is not accurate and largely invariant [see He et al., DOI: 10.1080/02786826.2019.1696015; Tryner et al., DOI: 10.1016/j.jaerosci.2020.105654; and Ouimette et al., DOI: 10.5194/amt-2021-170]. Tryner et al. show that PM1, PM2.5, and PM10 mass fractions calculated from the particle count data reported by the PMS5003 are not consistent with the PM1, PM2.5, and PM10 mass concentrations reported directly by the sensor [DOI:

10.1016/j.jaerosci.2020.105654, Figure 6]. Wallace et al. [DOI: 10.1016/j.atmosenv.2021.118432] also note this discrepancy in their Appendix ("Apparently, the hidden algorithms of the Plantower approach assign values to measurements that in some way depart from using the mass calculated from the numbers of particles in the size categories.").

Response 7:

Thank you for this helpful response and clarifying our misunderstanding of the operation of the Plantower sensor. In the revised document we clarified that PA reports particle count in bins as well as mass concentrations, both estimated from some unknown calibration applied to the scattering amplitude.

**Comment 8:**

1. Lines 74-75: "The CF ATM correction factor is derived from Beijing atmospheric conditions while CF 1 was derived from a lab study using symmetrical particles of a known size and is recommended for use in industrial settings (Zhou, 2016)." Where did the authors obtain this information? The PMS5003 manual that the authors cite does not contain any such statements

Response 8:

The original citation (Zhou, 2016) states "CF=1,   standard particle" and "[CF=atm,] under atmospheric environment" in the table in Appendix I.

We added "; Yang, personal communication, 2016" to the citation (Yang is from PlanTower, the manufacturer of the PMS5003 sensor, who we contacted for additional information when we initially began working with the PurpleAir monitors in 2016).

**Comment 9:**

1. Lines 76-77: "A recent study has developed a particle count correction factor using US-based sites which shows promise however that was not tested here (Wallace et al., 2021)." I'm not sure why the authors chose to give this correction approach special attention in this paragraph. The approach of Wallace et al. wasn't tested by the authors of this study and approaches to which the authors did compare their data aren't discussed until Lines 91-97. I'm not convinced that the approach of Wallace et al. is promising, either, since it seems to be based on flawed assumptions: that the PMS5003 measures particle counts and that the PM2.5 concentrations reported by the PMS5003 are calculated from the particle count data (see Comment 1).

Response 9:

We removed the discussion of the Wallace et al. paper on lines 76-77 (and lines 313-315).

**Comment 10:**

1. Lines 104-107: What was the domain for this analysis? Canada and the United States?

Response 10:

We added text stating that the domain covers Canada & USA.

**Comment 11:**

1. Lines 132-133: Why not download hourly PM2.5 concentration observations in the US from EPA AQS? There is a several-month lag between when the data are collected and when the data are available in AQS, but data in AQS have been QC'ed.

Response 11:

We did not use the AQS data as our data flows were setup for Airnow already, which provides both Canadian and US data, and the Canadian monitors would not be available from the EPA AQS.

**Comment 12:**

1. Lines 137-138: "We further removed several sites after viewing scatter plots of their valid PA and FEM PM2.5 observations and determining the performance to be unsatisfactory relative to the other sites." What criterion/criteria was/were used to determine that performance was unsatisfactory? Pearson correlation < 50%? The answer is not clear based on the current phrasing of this sentence and the one that follows.

Response 12:

We replaced "determining the performance to be unsatisfactory relative to the other sites." with "observing a non-linear relationship quite different from other sites."

**Comment 13:**

1. Lines 141-143: "PA RH values were restricted to the range 30%–70% (any values above/below this were set to 30% or 70%, respectively) as these values are near the efflorescence and deliquescence points typical of fine particulate matter (Parsons et al. 2004, Davis et al. 2015). Corrections utilizing RH tended to overcorrect observations at these extreme RH values." (a) Please specify the fraction of 1-hour average data points that were affected by this restriction. (b) What was the justification for modifying RH values below 30%? I don't recall seeing evidence in the literature that low RH values are a problem. Did the authors do a sensitivity analysis to see whether and how their results were affected by leaving in RH values below 30%? (c) It's unclear whether the authors decided to replace RH values above 70% based on prior results reported in the literature or based on the results of their own modeling. If this decision was based on prior results from the literature, the authors should cite relevant studies. If this decision was based on the authors' own modeling, it would be nice to see these results presented in the form of a sensitivity analysis.

Response 13:

a) We specified the fraction of hours where RH was missing or outside of 30% - 70% right before Table 1 in the revised paper.

b) We stated in the revised paper that the Barkjohn et al. study uses 24-hour average data which would tend to keep the average RH within the 30-70% range.

At extreme rh values the correction models can have an exponentially increasing impact on the raw data. We plotted all of our concentration data in a scatter (PA-FEM error on the y, RH on the x) and at RH outside of 30-70% the corrected PA error increased noticeably.

c) We added a figure of all of our concentration data in a scatter (PA-FEM bias on the y, RH on the x) to the SI and discussed this in the revised paper at the end of section 2.2.

**Comment 14:**

   1. Line 145: "Solar radiation impacts were too difficult to estimate given the variations in siting at each of the locations." I don't think the lack of correction for solar radiation is a big concern. The temperature and RH reported by the PurpleAir are biased high and low, respectively, even when the PurpleAir is installed indoors, due to heat generated by the electronics, so adjusting for solar   radiation would still not eliminate the bias.

Response 14:

This is correct, however, the heat from the electronics would be more consistent between sites and therefore easier to model and correct for. The fact that there are large temperature variations as well based on differences in siting makes it nearly impossible to correct for generally.

**Comment 15:**

   1. Lines 190-192: How did the authors assign sites to the training and testing datasets? Was this assignment done randomly? Or did the authors try to make sure the full range of geographic areas, climates, and seasons were represented in each dataset?

Response 15:

We added "Training/testing sites were randomly selected then adjusted (again randomly) to ensure representativeness across geographic areas and concentration ranges." to this section in the revised document.

**Comment 16:**

2.      Line 206: "increases" should be "increased".

Response 16:

We replaced "increases" with "increased" in the revised document.

**Comment 17:**

   1. Line 208: "The mean testing site concentrations had similar ranges for matching monitor types between the two data sets…" I don't understand what the authors mean by this. First the authors refer to "testing site concentrations", but then they refer to "the two data sets". Are the two data sets testing and training? FEM and PurpleAir?  What were the

"matching monitor types"? Does this phrase refer to the type of FEM monitor (e.g., beta attenuation) or to FEM vs. PurpleAir?

Response 17:

We replaced "The mean testing site concentrations had similar ranges for matching monitor types between the two data sets" with "Both the FEM and PA monitors had similar ranges of site median concentrations between the testing and training data sets"

**Comment 18:**

1. Figure 2: "Distributions of the Federal Equivalent Method (FEM) and PurpleAir training/testing sites median PM2.5 concentrations (µg m-3) at dry (0%–33%), moderate (34%–66%) and humid (67%–100%) relative humidity (RH) groupings." (a) How was each site categorised as dry, moderate, or humid? Based on the mean or median hourly RH at the site over the full data collection period? (b) Have the authors considered making the groups 0-30%, 30-70%, and 70-100% since they chose to modify RH values in the 0-30% and 70-100% ranges?

Response 18:

a) Each site was not categorized by their RH profile. Instead, the median values were calculated for every site for each of the "dry", "moderate", and "wet" hours. For each site, the median of the "dry" hours was calculated along with that for the "moderate" and "wet" hours.

We added the following summary of % of hours in each group to make this clearer.

"For the training data, 23% of the hourly observations were classified as "Dry" (RH ≤ 33%), 56% were classified as "Moderate" (30% < RH < 70%), and 21% were classified as "Humid" (RH ≥ 70%). The testing data were similar with 25%, 55%, and 19% of the hours classified as dry/moderate/humid, respectively.

b) We believe this small change would not significantly impact the results. We chose equal 33% groupings instead of 30%,40%,30%.

**Comment 19:**

1. Figure 3: Why were hours where the FEM AQHI+ was equal to 1 removed? For what fraction of hours was the FEM AQHI+ equal to 1? Most of them, right, considering the medians in Figure 2?

Response 19:

We clarified in the caption and the preceding text that these concentrations make up the bulk of the observations (skewing the results) and are unimportant from a health and management perspective. See also summary of the observation data presented in the response to Referee #2, comment 1.

**Comment 20:**

1. Table 2: Model 1 is similar to the model fit by Barkjohn et al., but with slightly different coefficients. Did the authors fit Model 1 to PurpleAir data that had been adjusted to replace RH values < 30% with 30% and RH values > 70% with 70%? If so, did the authors also try fitting Model 1 to a dataset without adjusted RH values? It would be interesting to see how much the coefficients fit using the dataset from this study differ from the coefficients fit by Barkjohn et al.

2. Table 2: Did the authors test Model 7, which was fit by Barkjohn et al., using the dataset in which RH values < 30% had been replaced with 30% and RH values > 70% had been replaced with 70%? If so, I'm not sure that's a fair test of this model because it wasn't fit using such adjusted RH values.

Response 20:

Yes, all of our models were fit using the truncated RH data. This improved the fit of all our models utilizing RH. We have clarified this in the text by adding "Truncating the RH data to 30 % - 70 % consistently improved the performance of RH-based models." to section 2.3

See comment #13. Barkjohn et al. used 24-hour averages to fit their data which effectively truncates the RH to ~30-70 in most locations. Therefore Model 1 would be quite comparable to that from Barkjohn et al.

**Comment 21:**

1. Figure 4: The color scheme used here was confusing to me. My initial reaction was that overestimates should be red and underestimates should be blue. Did the authors choose red for underestimates because they view the PurpleAir incorrectly underestimating the AQHI+ value, and therefore failing to alert the public to the true extent of the health hazard posed by air pollution, to be the worse outcome?

Response 21:

The colour scale (purple to orange) is not meant to display preference to bias either way, as low and high bias are both bad. The grey/white colour is the desired value. Orange and purple are easier to differentiate for most types of colour blindness (see "Accessible palettes" at https://davidmathlogic.com/colorblind/).

**Comment 22:**

2. Lines 256-264: Did the authors consider interpreting their results using the performance targets proposed in the US EPA Air Sensor Guidebook [document ID EPA/600/R-14/159]? In Section 5, the guidebook suggests precision and bias error < 50% for educational and informational purposes, < 30% for hotspot identification and characterisation or personal exposure monitoring, and < 20% for supplemental monitoring.

Response 22:

This guidebook is outdated (released June 2014) - a more recent set of protocols was defined in EPA Document ID 350785 (*Performance Testing Protocols, Metrics, and Target Values for Fine*

*Particulate Matter Air Sensors: Use in Ambient, Outdoor, Fixed Site, Non-Regulatory Supplemental and Informational Monitoring Applications*) released February 2021. Table 3-1 defines metrics to consider, but thresholds are not provided.

We added the following discussion of this to section 2.4.

"Duvall   et al. (2021) outline several key metrics to consider for small sensor performance: precision, bias and error, linearity, effects of RH and temperature, sensor drift, and accuracy at high concentrations. Evaluating precision is not viable in this study given that many sites only had a single PA installed. We will evaluate bias, error, and linearity through our analysis, as well as the effects of RH. We found temperature impacts to be minimal for our dataset, especially when the impacts of RH were already considered. Sensor drift is outside of the scope of our study, and accuracy at high concentrations is less of a concern given our use of the AQHI+ scale and focusing on the moderate to high concentrations."

**Comment 23:**

1. Lines 299-300: "The Canadian AQHI+ system was useful as a framework for evaluating correction models across a range of concentrations infrequent high values or numerous low values can skew performance statistics when evaluating the full range at once." Is there a word missing here? Was this supposed to be two sentences?

Response 23:

We inserted ", as" between "concentrations" and "infrequent".

**Comment 24:**

2.       Lines 301-304: The names Model 2, Model 7, etc. are not very informative to a reader who is not looking at Table 2. It would be helpful to describe the key features of Models 2 and 7 here. Please also explain that models using RH as a predictor were fit and evaluated after replacing RH values below 30% with 30% and RH values above 70% with 70%. I don't think it's necessary to note Models 3 through 6 in this paragraph.

Response 24:

We added clarification to this paragraph briefly describing each model. Ie. "... Model 7, the humidity multiple regression from Barkjohn et al.". We also removed discussion on Models 3 through 6 here.

**Comment 25:**

1. Lines 305-306: "…the average performance across the testing sites and over time was evaluated here; performance at colocation sites and across time was not the same." Why did the authors choose to focus on the average performance across testing sites and over time? Were there any sites or times where Models 2 and 7 performed notably better or worse? If so, what were the notable features of these sites and times (weather, unique PM sources) and what do those features say about the advantages and limitations of Models 2 and 7?

Response 25:

It was our objective to "provide an optimised correction model for north american PA sensors" - without further data/effort a regional/seasonal specific model cannot be developed so a more general model was required, especially for utilizing the PA sensors to support AQ management.

We added a figure to the SI breaking down the mean AQHI+ bias figure into individual testing sites and referenced it at the start of section 3.2 as well as in section 4. Site 37 (in Oregon) had the most notable improvement after correcting, sites 43-45 (in Nebraska, Alaska, and California, respectively) had the most notable reduction in performance after correcting. All of these sites are in considerably different geographic locations - outlining the difficulty in developing a regional based correction model without additional colocation sites.

**Comment 26:**

1. Lines 313-315: "…the improved particle bin correction factor proposed by Wallace et al. (2021) should be implemented for these sites…" See Comment 3. I don't understand why the authors repeatedly refer the correction approach proposed by Wallace et al. I don't think it's a good approach or particularly relevant to the work presented here.

Response 26:

See response # 9. Removed the Wallace et al. discussion.

**Referee #2 Response Letter**

Thank you for the time you put into reviewing our manuscript and the very useful and helpful feedback which has led to improvements in the paper. Please see our following responses and proposed alterations which we believe will resolve your individual comments.

**Comment 1:**

1. It seems like you a missing a summary of the dataset. What is the range of hourly concentrations? How many points are there per AQHI category? Did you see nonlinearity in the high concentration data (https://doi.org/10.1111/ina.12621)?

Response 1:

We added the following text and a new figure after Figure 1 in the results section.

"Hourly concentrations of PM2.5 ranged between $0 - 837$ µg m-3 and $0 - 986$ ug/m3 across all sites during this period for PA and FEM monitors, respectively (Figure 2). PA monitors at most sites tended to be within a factor of 2 of FEM, typically biased higher. For most sites this bias appears to be linear as concentrations increase. PA PM2.5 concentrations across all sites were categorised as "Low AQHI+" ($0 - 30$ ug/m3) for 91.1% of observations, "Moderate AQHI+" ($30 - 60$ ug/m3) for 7.7%, "High AQHI+" ($60 - 100$ ug/m3) for 0.7% and "Very High AQHI+" ($100+$ ug/m3) for 0.6% of observations. In the same order for the FEM monitors at all sites: 97.5%, 1.9%, 0.3% and 0.4% of observations were in the four AQHI+ categories."

**Comment 2:**

2.     Model 5 should also be applied to the cf_atm data. This is the AQ&U equation from the PurpleAir map. Although a cf isn't listed on the PurpleAir map you can check which cf it is by checking the calculation at a high concentration site (since we know the "raw" outdoor data is cf_atm) (I did this today Feb 8th and it seems to still be applied to the cf_atm data). The Kelly paper was published in 2017 long before PurpleAir flipped the labels to reflect Plantower's labels. Further confirmation: cf_atm is used in this equation in this recent study: https://amt.copernicus.org/articles/14/4617/2021/

Response 2:

Changed "Kelly et al. (2019)" to "Kelly et al. (2017)" in Table 2 and swapped CF_1 for CF_atm. Regenerated figures/results with this adjusted model. This model now performs comparably to the Barkjohn et al. 2020 model. Adjusted the results and conclusions to reflect this (Model 5 now discussed in favour of Model 8).

Comment 2:

1. Did you consider whether RH and PM5 are correlated at hourly averages in your dataset? In multiple linear regression independent variables should be independent.

Response 2:

Yes, they were not strongly correlated at most locations. site-wide Pearsons correlation coefficients ranged from -0.13 to 0.48 with a median correlation of 0.18.

**Comment 3:**

2.     Lines 75-77: The "CF 1" data were found to correlate better with FEM observations in our data set. A recent study has developed a particle count correction factor using US based sites which shows promise however that was not tested here (Wallace et al., 2021)." This belongs in results/discussion not introduction. Sharing the correlations for the CF_1 vs CF_atm data would be helpful.

Response 3:

Removed discussion of Wallace et al. in response to another comment from Referee 1.

**Comment 4:**

3.     Lines 85-87: "In addition, we and others have found the PA temperature observations to be biased high (and in turn RH biased low) because of internal heat produced by the electronics as well as incoming solar radiation (which has varying impacts depending on the physical location and placement of each monitor)" your results should go in the results & discussion. It would be good to include citations here of past work showing warmer and dryer (e.g. https://doi.org/10.3390/s20174796, https://doi.org/10.1080/02786826.2019.1623863)

Response 4:

We altered the phrasing here so it was more a statement of potential error and less a comment on results we have noticed as considering biases in T and RH were not a purpose of our study.

**Comment 5:**

4.       Update the Barkjohn 2020 AMTD preprint article to the final published AMT article.

Response 5:

We updated the reference and the intext citations.

**Comment 6:**

1. Lines 104-109: Can you clarify did you download all nearby sensors or only sensors that were labeled as outdoor sensors?

Response 6:

We clarified in the revised text that we only selected sensors labelled as outdoor sensors.

**Comment 7:**

2.       Line 125: a) How many months were flagged as invalid for temperature and RH? And what is the fraction of months removed from sensors where this is an issue? It would be good to understand the break down by sensor to understand are these sensors that were labeled as outdoor but are always running indoors or are they just being brought indoors for a month and then returned to the outdoors? b) Did you check whether this worked correctly with sensors that were marked by the user as indoor sensors?

Response 7:

a) Added the following paragraph to the beginning of the results detailing the % of data removed by each QA/QC step and the number of sites affected by each.

"The colocation site selection metric we used detected 86 potential colocation sites during this period in Canada and the United States. All sites had missing data, five sites had PA sensors with manually flagged invalid data, 65 had months where the temperature or RH were deemed too invariable to be outdoors, 67 had hours flagged invalid from differences between the A and B sensors within the PAs, and six sites had monitors with less than two months of valid data. Across  all of these sites, 40.1% of the PM2.5 observations were missing (either from the FEM or a PA), <0.0001% were manually flagged as invalid, 3% were flagged as months where the PA was likely indoors, 2.3% were flagged by our PA A/B sensor comparison, and 1.3% were removed from PA monitors with less than two months of valid data."

b) We did not - but we confirmed it worked at sites we knew were outdoors.

**Comment 8:**

3.       Line 128: Would you need to provide someone with a cut off level for a Hampel identifier (or other input variable) for them to recreate this method? Did you use a software package to complete this analysis?

Response 8:

Added to the manuscript that the standard cuttoff was used (outliers exist >3 median absolute differences from the median).

See response 11

**Comment 9:**

1. How much data was removed with each QC step?

2. Line 129: How many months had <72 hours of data?

3. Line 131: How many hours or what percent were removed by manual inspection?

4. Line 133: How many sensors had <2 months of data?

Response 9:

See response 7

**Comment 10:**

5.    Line 128: Can you clarify what you mean by "3 units"

Response 10:

Added "(ie. 3 °C for temperature or 3% for RH)" to this sentence.

**Comment 11:**

6.    What software was used for this analysis?

Response 11:

Added a statement of the software used (R) plus several key R packages.

**Comment 12:**

7.    Line 132: Why not use the QC'd data from AQS?

Response 12:

Our data flows were setup for Airnow already, as the Canadian monitors would not be available from the AQS site.

**Comment 13:**

8.    "Sites with multiple colocated PA monitors were averaged together to produce a single data record for each site after flagging and removing any invalid data." Did you consider the precision of PurpleAir sensors in places where multiple sensors were close by?

Response 13:

They correlate well with each other and tend to be within a few µg m$^{-3}$ from my experience, but no we did not analyse that specifically for this dataset.

**Comment 14:**

9.      Line 149-150: Could you clarify what you mean here? "Piecewise models which were built starting from the second segment tended to perform better in the mid-range PM2.5 concentrations than those built starting from the first segment."

Response 14:

Normally piecewise models are built starting from the first segment, then you fit the next segment from the end of the first, and so on. Since we wanted to focus on the moderate-high ranges we tried fitting this middle range first, then fitting the first and last segments to the ends of the middle.

We clarified this in the revised text (adding the underlined text in the following sentence)

"Piecewise models which were built starting from the second segment (fitting the mid-range data first) tended to perform better in the mid-range PM2.5 concentrations than those built starting from the first segment (fitting the low-range data first)"

**Comment 15:**

1.   Line 175: The US EPA performance targets for PM2.5 sensors may be valuable for this work ("Performance Testing Protocols, Metrics, and Target Values for Fine Particulate Matter Air Sensors: Use in Ambient, Outdoor, Fixed Site, Non-Regulatory Supplemental and Informational Monitoring Applications"https://cfpub.epa.gov/si/si_public_record_Report.cfm?dirEntryId=350785&Lab=CEMM)

Response 15:

We added the following discussion of this to the paper in section 2.4.

"Duvall et al. (2021) outline several key metrics to consider for small sensor performance: precision, bias and error, linearity, effects of RH and temperature, sensor drift, and accuracy at high concentrations. Evaluating precision is not viable in this study given that many sites only had a single PA installed. We will evaluate bias, error, and linearity through our analysis, as well as the effects of RH. We found temperature impacts to be minimal for our dataset, especially when the impacts of RH were already considered. Sensor drift is outside of the scope of our study, and accuracy at high concentrations is less of a concern given our use of the AQHI+ scale and focusing on the moderate to high concentrations."

**Comment 16:**

2.      Line 175: Could you provide calculations for these metrics here or in the SI? (since there has been some discrepancy on calculation method especially for RMSE in the sensor literature)

Response 16:

We added these metrics to the revised manuscript in section 2.4 (Eq. 6 – 11).

**Comment 17:**

1. Figure 2 only shows the site medians. Could you also add a figure showing the full dataset of hourly points? Maybe a scatter plot of hourly FEM PM2.5 vs PurpleAir PM2.5?

Response 17:

See response 1

**Comment 18:**

Table 1: Did you try looking for these in AirNow Tech by method_code? (https://aqs.epa.gov/aqsweb/documents/codetables/methods_all.html)

Response 18:

Yes, we pulled our information on the US sensors from the AQS database.

**Comment 19:**

2. Figure 3: Did you consider whether just the 3 known types were significantly different?

Response 19:

The 3 known types were not significantly different; we tested this, and the p value went from 0.15 to 0.11 after removing the unknown category.

**Comment 20:**

3. Table 2: It may be clearer to use letters to represent the sources so that they are not easily confused with the model numbers.

Response 20:

We modified the table to do this.

**Comment 21:**

4. The conclusion would be easier to interpret if when referring to the models by number it also described the model type.

Response 21:

We added clarification to the conclusion briefly describing each model. I.e. "... Model 7, the multiple (RH) regression from Barkjohn et al."

**Comment 22:**

5.        Line 305: "It should be noted that the average performance across the testing sites and over time was evaluated here; performance at colocation sites and across time was not the same." Did you consider whether there were regional or factors that could explain this?

Response 22:

Yes, one of our future recommendations is to develop a seasonally and regionally specific model as more data/sites become available.

**Referee #3 Response Letter**

Thank you for the time you put into reviewing our manuscript and the very useful and helpful feedback which has led to improvements in the paper. Please see our following responses and proposed alterations which we believe will resolve your individual comments.

**Comment 1:**

Line 123: Please state the reasoning behind your choice of 5ug/m^3 as the absolute error cut-off for identifying failures in either sensor. Is this a recommendation from PA?

Response 1:

We added a statement that this method and cutoff was derived from methods proposed by Barkjohn et al. 2021 and Tryner et al. 2020.

**Comment 2:**

Line 138: "The final set of colocation sites (47 in total) were then selected as those with at least half a year (4380 hours) of valid data from both PA and FEM and a minimum correlation of 50% for all valid hourly observations over the period of record." What is the reasoning for setting the minimum correlation to 50%? While this would remove any non-collocated sensors wouldn't this also possibly remove any poorly performing collocated sensors?

Response 2:

The minimum correlation of 50% was iteratively decided such that the sites that were clearly poor performers (i.e. those potentially not colocated) were removed, while known colocation sites were not. This method is not perfect, but a cutoff needed to be decided given the automated nature of our approach.

**Comment 3:**

Line 148: "A temperature term was also tested; however, its impact was found to be minimal." Does this statement refer to the pooled together training dataset? Was this ever tested for individual sites?

Response 3:

Anytime we added temperature to a model it performed worse than a model with just RH or both combined (in general and at a few sites we initially worked with on the individual level). For

clarification, in the revised paper we added "and given the high correlation between temp and rh, rh was selected as the more important term" to this sentence.

**Comment 4:**

Line 159: In equation 3 please clarify that the correction factors of a and b for PM2.5 < x are different than the factors a and b for x< PM2.5< x2.

Response 4:

We modified the table to reflect this.

**Comment 5:**

Line 191: How were the training and testing sites divided up? Rather than dividing by site did you consider dividing by time (randomly dividing to ensure similar conditions between testing and training datasets)? Please clarify the reasoning between choosing 32 training sites and 15 testing sites (~2/3 training).

Response 5:

We added the following to this section: "Training /testing sites were randomly selected then adjusted (again randomly) to ensure representativeness across geographic areas and concentration ranges."

Yes, we considered dividing by time, but given the seasonal differences that can occur (and the variation in this between locations) we felt this would either result in 1) one of the datasets (testing or training) having a disproportional amount of episodic concentrations (winter inversions, wildfire smoke, etc) or 2) the two datasets being highly dependent on each other (if you took every third day as test for example).

We initially had more testing sites than training sites, but we found that the other way (what we presented here) was more common in the studies we cited that had a large number of colocation sites.

**Comment 6:**

Line 246: "Further comparisons were only made on Models 1, 2, 7, and 8 as they showed the best performance here." It would be informative to include the results from the other models in a supplementary information section.

Response 6:

We added two figures and a table to the SI document with the results from the remaining models and added a reference to them at the start of the results section.

**Comment 7:**

Line 294: "The concentrations of PM2.5 reported from the PA monitors were biased high compared to the FEM monitors at most colocation sites, especially for the lower concentration

range." While you evaluated model bias at different PM2.5 concentrations did you consider looking at the bias over a RH range?

Response 7:

At extreme RH values the correction models can have an exponentially increasing impact on the raw data. We plotted all of our concentration data in a scatter (PA-FEM error on the y, RH on the x) and at RH outside of 30-70% the bias in PA changed noticeably (see new figure in SI).

**Comment 8:**

Line 324: For scenarios where testing models on individual locations is not an option, such as applying a correction in an area without a nearby PA-FEM colocation site, we recommend using our Model 2." Rather than use a model that has been trained on a variety of locations/conditions and therefore is pretty generalized, would it not be more prudent to use a model that has been trained on conditions similar to those you expect to encounter?

Response 8:

Yes, but this statement is for the scenario where those data are not available. At the moment there are many correction formulae that have been developed, several of our tested ones come directly from the PurpleAir map and do not perform as well. We posit that our Model 2 would be better to use than these other models for general application across many locations (like you would on a mapping platform).

---

## Author Response (AR2)

[revised manuscript text omitted]

---

## Author Response (AR3)

**Development and Evaluation of Correction Models for a Low-Cost Fine Particulate Matter Monitor: Final Author Response**

Brayden Nilson[12], Peter L. Jackson[1], Corinne L. Schiller[12], Matthew T. Parsons[2]

[1]Department of Geography, Earth and Environmental Sciences, University of Northern British Columbia, Prince George, V2N 4Z9, Canada
[2]Air Quality Science – West, Meteorological Service of Canada, Environment and Climate Change Canada, Vancouver, V6C 3S5, Canada

**Response to Referee #2 Report**

1. Line 23: missing 2.5 subscript.

Subscripted 2.5 here.

2. Line 37: …ensures FEM observations are equivalent to reference monitoring methods [at 24-hr averages]. -suggest inserting end of sentence. At least in the US FEMs are only evaluated against 24-hr FRM measurements. There may be more uncertainty in 1-hr measurements from FEMs.

Add "at 24-hr averages" to this sentence.

3. Line 41-43: FEM monitors can be one of multiple potential sensor [measurement] types; each of which measures PM2.5 using different techniques (such as beta attenuation, gravimetric, and/or broadband spectroscopy). These sensors [FEMs] are rigorously tested and compared with 24-hour average reference measurements from Federal Reference Monitors (FRM) to ensure comparability. -I think you started referring to FEMs as sensors here. Recommend not using sensors to refer to FEMs for clarity.

Clarified throughout the document that we are referring to the internal observation device when we say "sensor" (ie. Plantower PMS5003 for the purpleair) and to the actual unit (including sensors, wifi cards, dataloggers, etc) when we say "monitor". So for FEM we say monitor for the most part, but here we are discussing the sensor types between monitors.

4. In some places you refer to PA sensors and in other places PA monitors. Recommend being consistent with terms and sticking with PA sensors throughout.

See above comment – clarified this in the paper in multiple locations.

5. Line 115: "of each other" I think you mean "of an FEM"

Replaced "of each other" with "of a FEM"

6. Line 151: It would be helpful to put these plots in the SI. Do you think this is due to differences in sensor performance? Or due to not a true collocation/localized sources?

Added "These relationships were most likely the result of the PA not actually being colocated outdoors at the site."

7. Eqn 6-8: Can you define all your variables (modi, obsi)

Defined each variable in these equations.

8. Figure 2: Impossible to see the dotted 1:1 line in the ~0-150 ug/m3 range. Consider using a different color (red?) so it is more visible.

Changed the lines to red here and drew them on top of the data (instead of behind).

9. Figure 2: Also I don't think you discussed general additive models in the text anywhere. It seems like it would be more straight forward to plot linear regression or one of the other equations 2-5 you plan to test. Why didn't you consider general additive models as one of your equations if you are using them as lines on this plot?

Added "used as opposed to a linear model to provide a general overview of site PA/FEM comparability". We found GAM's tended to overfit when provided multiple sites, so they were not used, but felt that they gave a good general representation site-by-site.

10. Line 256: "PA concentrations tended to be biased increasingly higher as humidity increased; this was not the same case for the FEM monitors…" Do you have a figure or other analysis (e.g. correlation) documenting this?

This comment was directed at Figure 3 – rephrased it a bit to make that more clear that was the case.

11. Table 2: Model 5 needs to be updated to cf_atm

Updated model 5 in table 2.